An experience selecting quality features of apps for people with disabilities using abductive approach to explanatory theory generation

Larco Andres andres.larco@epn.edu.ec 1
Montenegro Carlos 1
Yanez Cesar 1
Luján-Mora Sergio 2
1 Departamento de Informática y Ciencias de la Computación, Escuela Politécnica Nacional , Quito , Pichincha , Ecuador
2 Department of Software and Computing Systems, Universidad de Alicante , Alicante , Spain
Aljawarneh Shadi
Electronic publication date: 2021 Aug 2
Publication date: 2021
Volume: 7
Electronic Location ID: e595
Received 2021 Mar 19; Accepted 2021 May 24
Copyright: ©2021 Larco et al.
Copyright year: 2021
Copyright holder: Larco et al.
License: This is an open access article distributed under the terms of the Creative Commons Attribution License, which permits unrestricted use, distribution, reproduction and adaptation in any medium and for any purpose provided that it is properly attributed. For attribution, the original author(s), title, publication source (PeerJ Computer Science) and either DOI or URL of the article must be cited.
License URL: https://creativecommons.org/licenses/by/4.0/

Keywords: Abduction, Apps quality, Data mining, Explanatory theory generation, People with disabilities

Funding: The authors received no funding for this work.

==============================
This study determines one of the most relevant quality factors of apps for people with disabilities utilizing the abductive approach to the generation of an explanatory theory. First, the abductive approach was concerned with the results’ description, established by the apps’ quality assessment, using the Mobile App Rating Scale (MARS) tool. However, because of the restrictions of MARS outputs, the identification of critical quality factors could not be established, requiring the search for an answer for a new rule. Finally, the explanation of the case (the last component of the abductive approach) to test the rule’s new hypothesis. This problem was solved by applying a new quantitative model, compounding data mining techniques, which identified MARS’ most relevant quality items. Hence, this research defines a much-needed theoretical and practical tool for academics and also practitioners. Academics can experiment utilizing the abduction reasoning procedure as an alternative to achieve positivism in research. This study is a first attempt to improve the MARS tool, aiming to provide specialists relevant data, reducing noise effects, accomplishing better predictive results to enhance their investigations. Furthermore, it offers a concise quality assessment of disability-related apps.

Introduction

There are several definitions of theory. One, established by Sjøberg et al. (2008), depends on philosophical and practical issues and the field of study. However, Corley & Gioia (2011) offer a more straightforward definition, a statement of theories and their interrelationships that shows how and why an exceptional event occurs. But Horváth’s (2016) theory explains the concepts and facts in a given context, matching ideas and events logically based on their meaning, which similarly indicates the limits of the theory, facilitates the applicability and permits the recognition of new hypotheses to cover a broader field.

According to Wacker (1998), a theory contains four components: definitions, domain, relationships, and prediction. Aliseda (2006) suggests that discovering an idea towards a new theory involves a complicated process starting with the initial conception through to an acceptable conclusion, thereby forming a new theory.

Nevertheless, according to Philipsen (2018), knowledge production divides into three specific categories: discovery, problem/domain definition, initial concepts, and also the context of justification, theories testing, as well as hypotheses enhancement. Researchers understand reason more readily than discovery.

The justification uses three interrelated types of reasoning. Ngwenyama (2014) inferred that the deduction probably assists in suggesting logical implications of rules to develop experiments for observation as well as testing. Induction enables the scientist to deduce general rules from the monitoring of consistencies in phenomena behavior. Abduction is primarily an inference of an explanation of the views analyzed by Ngwenyama (2014) and Flach & Kakas (2000). Lastly, Kapitan (1992) suggests abduction is the procedure of generating theories and also developing some of them; reduction extracts their testable effects while induction assesses them.

The primordial feature of the rule is the variable individual measurable property of a process being observed. Feature selection helps understand data, which reduces calculus requirement and the effect of dimensionality while additionally improving the predictor performance. Consequently, the relevance of the attribute option is to select a subset of input variables that can describe data, minimizing noise or irrelevant variables, and yet offer more accurate predictive results (Guyon & Elisseeff, 2003; Chandrashekar & Sahin, 2014; Saeys, Inza & Larranaga, 2007).

An exploratory approach to related contributions shows little research has been conducted on the fore-mentioned topic. Thus, new experiences are required to amplify the application domain options and corroborating the relatively new abductive approach. The current study uses the abductive process to create theories to improve the formal application of the Mobile App Rating Scale (MARS) results (Stoyanov et al., 2015), the tool used to evaluate apps quality for people with disabilities. The research estimates the evaluation of the tool’s external consistency because the MARS tool results are unable to be used for identifying relevant and unique variables that represent the quality factors. The objective of this work is to simplify the MARS tool to increase its performance without losing the quality of the evaluation. As far as we know, this study is the first attempt to improve the MARS tool. The principal contribution is to provide specialists relevant data, reducing noise effects, and accomplishing better predictive results to enhance their investigations.

The structure of the current study is: ‘Background and Related Work’ presents background and related work that includes the definitions of abductive reasoning and its applications; ‘Research Approach’ contains the research approach, in three stages: the result, the rule, and the case; ‘Discussion’ involves discussions of the results obtained, while ‘Conclusions and Future Work’ presents various conclusions.

Background and related work

Philipsen (2018) and Ngwenyama (2014) establish the distinctions between deductive, inductive, and also abductive reasoning with the connections between the entities; rule, case, and result. These three forms of scientific thinking are presented in Fig. 1.

Figure 1 Inference forms.

Adapted by permission from Springer: Springer Nature, Collaborative Research Design, Philipsen (2018).

The abductive reasoning process addresses the situation where the findings differ from the theory’s anticipated result, which guides the research study. The starting point coincides with that of induction but is concerned with the search for an explanation of the results, which are complex to explain applying the initial guiding theory. The search for reason demands the need for a new hypothesis, leading to the specific investigated case (Philipsen, 2018). Aliseda (2006) assumes that abduction in the scientific sense refers to empirical progress, pragmatism, and epistemic change.

O’Reilly (2016) and Flach & Kakas (2000) concluded that abduction is the only logical operation that permits new ideas. In testing theory, abduction develops phases of the knowledge-production process. New explanations will likely arise where there is a requirement to solve an anomaly and discover new methods of explaining the particular empirical phenomenon (Philipsen, 2018). Philipsen et al. established the research gaps, and the results are considered vital factors for identifying inconsistencies.

Aliseda (2017) indicated logical abduction is relevant regarding issues of scientific explanation. More recently, logical abduction found a place in computationally oriented theories of belief change in Artificially Intelligence. Olsen & Gjerding (2019) investigated the notion of abduction related to and can be applied in a scientific research study. Furthermore, it showed the most necessary treatments of abduction in modern times, and it tried to define various processing modalities, both as an autonomous research strategy and inference type, and in relation and contrast to induction and deduction.

According to Zelechowska, Zyluk & Urbański (2020), abduction is a type of complex reasoning carried out to make sense of unusual or ambiguous phenomena or fill the gaps in our beliefs. Despite the ubiquity of abduction in professional and everyday problem-solving processes, little empirical research was dedicated to investigating this kind of reasoning. Most of them concentrated on products of abduction-abductive hypotheses. Rapanta (2018) explored abductive reasoning as the most appropriate for students’ arguments to emerge in a class discussion. Abductive reasoning embraces the concept of plausibility and defeasibility of both the premises and the conclusion.

Mitchell (2018) posits that pragmatism supports using various research techniques, which a continual cycle of inductive, deductive, and when proper, abductive reasoning creates practical knowledge and works as a rationale for a rigorous research study. Abductive reasoning was essential for explaining empirical phenomena relating to competition, primarily how the top United Kingdom and German multinationals developed various strategies for outsourcing. Moreover, applying different methods can lead to research and succeeding management choices that reflect both the interplay of social and scientific elements of the world today. The work of Mitchell (2018) is focused on the strategies for outsourcing. This is in contrast to our work, as we concentrate on app quality.

Fariha & Meliou (2019) present the idea of an abduction-ready database, which precomputes semantic features and related statistics, allowing semantic similarity-aware query intent discovery (SQUID) to achieve real-time performance. Also, an extensive empirical assessment was provided on three real-world datasets, consisting of user-intent case studies, demonstrating that SQUID is efficient and effective and outperforms machine learning techniques. In contrast to our research, there is no assessment of the quality of the apps for data processing.

Ganesan et al. (2019) propose a probabilistic abductive reasoning method that enhances an existing rule-based Intrusion Detection Systems (IDS) to detect these evolved attacks by predicting rule conditions that are likely to occur and able to generate new snort rules when provided with seed rule to reduce the concern on experts to update them constantly. This is in contrast to our study, as we focused on feature assessment of apps.

Bhagavatula et al. (2020) present the initial study that research the viability of language-based abductive reasoning. Also, conceptualize and introduce Abductive Natural Language Inference (ANLI)—a novel task focused on abductive reasoning in narrative contexts. The task is formulated as a multiple-choice question answering problem. Additionally, introduced Abductive Natural Language Generation (ANLG)—a novel task that requires machines to generate plausible hypotheses for given observations. In our study, we only focus on optimizing the MARS.

A review of related work shows that the abductive process has been used in various forms and specialties related to Information Systems (IS) and Information Technology(IT). The abduction process has more theoretical development than practical in relation to the integration of abduction and induction (Flach & Kakas, 2000). Other works consider digital interaction’s abduction paradigm to be a research paradigm (Patokorpi, 2006; Patokorpi & Ahvenainen, 2009).

To solve the problems of single-case research, the approach based on the systematic combination in an abductive logic was implemented to improve theory development (Dubois & Gadde, 2002). Flach & Kakas (2000) contributions to abductive reasoning included logic programming, machine learning, and artificial intelligence.

The theory development in software engineering combines mainly inductive and abductive aspects, which may initiate from both the practical and theoretical perspectives. For example, in the related work, the abductive approach is applied to software requirements (d’Avila Garcez et al., 2003) and software testing (Angius, 2013). Osei-Bryson & Ngwenyama (2013) explore and illustrate the use of IT in IS research to assist researchers in the testing of theories and developments through mining techniques (Osei-Bryson & Ngwenyama, 2014b), decision trees (Osei-Bryson & Ngwenyama, 2011), or logical foundations (Ngwenyama, 2014).

Research approach

The result: data collection and evaluation

This research study contains a group of apps that focuses on the needs of people with an intellectual disability who were assessed applying a specialized evaluation tool. The data collection contains some components and processes, which are following described.

The MARS tool

As Holzinger et al. (2008) determine, metric-based reference points are significant for quantifying software program usability, particularly for specific end-user groups. One of the principal qualities is its usability, as it is an indispensable feature of all software. It is even more crucial in apps created for a large range of users. Additionally, the requirements of people with disabilities are ruled out in the basic needs extraction procedure (Guerrero & Vega, 2018). The fundamental elements of software-based clinical systems are; software apps measurement, quality assurance, and end-user satisfaction (Ahamed et al., 2012).

MARS is rated as an outstanding quality tool for efficient use for mobile health apps, developed from a methodical literature search to determine apps quality criteria (Stoyanov et al., 2015). MARS is a well-established tool worldwide that has been consulted by 201,436 academics, cited by 526 researchers, and 78 tweets.

MARS scale assesses app quality on four dimensions, with similar grading to the Likert scale, e.g., “1. Inadequate” to “5. Excellent”, 18 questions and descriptors were used (Stoyanov et al., 2015):

• Engagement: entertainment, interest, customization, interactivity, and target group.

• Functionality: performance, ease of use, navigation, and gestural design.

• Aesthetics: layout, graphics, and visual appeal.

• Information: accuracy of app description, goals, quality of information, the quantity of information, visual information, and credibility.

Table 1 shows the most relevant studies that use MARS. Selected articles use MARS to evaluate different types of apps in the health field.

Table 1 Related investigations.

Reference	Summary	
Osei-Bryson & Ngwenyama (2013)	Evaluated 34 apps with MARS related to heart failure symptom monitoring and self-care management. Reviewed by the article authors.	
Osei-Bryson & Ngwenyama (2014b)	Assessed 23 iOS apps with MARS. The engagement category had the lowest score and highlights the lack of attractiveness. Reviewed by the article authors.	
Osei-Bryson & Ngwenyama (2011)	A group of 272 medication reminder apps was classified. Only ten apps were evaluated with MARS. Reviewed by the article authors.	
Holzinger et al. (2008)	The study analyzed asthma apps with the potential to promote patient’s self-management. Thirty-eight apps were evaluated. Reviewed by the article authors.	
Guerrero & Vega (2018)	Describes features of 40 apps which collect personal data and dietary behavior. 20 travel apps and 20 dietary apps were assessed with MARS. Reviewed by the article authors.	
Ahamed et al. (2012)	The study assessed features of apps that assist people to monitor Rheumatoid arthritis disease activity. 11 Android and 16 iOS apps were evaluated through MARS. Two independent reviewers.	
Liberati et al. (2009)	The study established the quality and sharpness of 58 apps for drink driving prevention. Reviewers not specified.	
Hutton et al. (2015)	Using MARS, 89 apps were assessed for diabetes management to see if they have enough quality to complement clinical care. Reviewed by three people.	
Straub & Gefen (2004)	Twelve mHealth apps that give the user behavioral and cognitive skills to manage insomnia were evaluated with MARS. Reviewed by two authors of the article.	
Heale & Twycross (2015)	Five iOS apps for self-managed balance rehabilitation for older adults were assessed with MARS. Reviewed by two authors of the article.	
Miao & Niu (2016)	Characteristics of 23 potential Drug-Drug Interaction apps were reviewed and evaluated with MARS. Reviewed by two testers per app.	
Dy & Brodley (2004)	Conducted a systematic review of apps related to epilepsy. Found and evaluated 20 apps with MARS focused on educating people about their condition. Reviewed by a research team.	
Law, Figueiredo & Jain (2004)	Ten people assessed 54 apps; the research contributes with new insights about how to use mHealth apps to assist cancer survivors’ physical exercise.	
Kohavi & John (1997)	An interdisciplinary team of clinicians, behavioral informatics, and public health reviewers trained in substance use disorders conducted a descriptive analysis of 74 apps using MARS.	
Bolón-Canedo, Sánchez-Maroño & Alonso-Betanzos (2016)	Participants were randomly assigned to interact with either the high behavior change technique app, or the low behavior change technique app using an iPad. Participants then completed a MARS questionnaire.	

Apps collection

In order to obtain accurate results, a relevant issue is the determination of the number of apps sampled and apps evaluated. Exploratory activity was performed doing a data compilation of web and mobile apps for people with disabilities using information from the year 2000 to 2020. Figure 2 present the available data and showing the exponential growth of the number of apps in the period. Besides, the data universe size suggests that a good selection is a census of a specified domain for specific users and downloaded in an “instant” period.

Figure 2 Number of web and mobile apps published between 2000 and 2020.

The present research makes use of the Preferred Reporting Items for Systematic Reviews and Meta-Analyses tool (PRISMA) (Liberati et al., 2009) o select the appropriate apps for testing in the MARS tool. PRISMA (Hutton et al., 2015) consists in a four-phase flowchart: identification, screening, eligibility, and inclusion. The apps were selected on different platforms. Table 2 illustrates the search and inclusion process conditions.

Table 2 Search process.

Search string	Inclusion criteria	Period	Web site, stores, and repositories	
Apps for life skills	Educational apps	Feb 2020-Oct 2020	Wikinclusion, Google Play Store, Apple App Store, PhET, Genmagic.org, Educaplanet, ARASAAC, Pictoaplicaciones, Juegos Infantiles Pum, Edujoy, CEDETi, Educación inclusiva ONCE, Proyecto DANE, Proyecto Comunica, pescAPPs, MyFirstApp, OWLIE BOO, Fundación Orange.	

The researchers chose the apps across four platforms: desktop 22.83%, web 33.45%, Android 22.12%, and iOS 21.59%.

Apps evaluation

There are three stakeholder groups to assess apps for people with disabilities: health specialists, software specialists, and final users. The initial approach used a sample of four apps which teachers, software testers, and children with disabilities evaluated. A group of 10 specialized teachers for people with disabilities, 15 children with special educational needs or intellectual disabilities, and five software testers used MARS to evaluate the apps.

The authors created a new Spanish version based on the MARS template. Still, due to the questionnaire’s size and complexity, it was necessary to adapt it for children with disabilities and test it. Although the results of the teachers’ and children’s evaluations of the apps were similar, the software testers’ evaluation shows some discrepancies, as shown in Fig. 3.

Figure 3 Joint assessment of four apps for people with disabilities.

The complete test series included a total of 1,125 apps after a PRISMA screening process deleted duplicates and non-available apps, having a result of 565 apps, where 123 iOS apps, 125 Android apps, 190 web apps, and 127 Windows apps to be evaluated with MARS. Two independent software testers performed the evaluation. Table 3 shows the devices used to complete the assessment.

Table 3 Devices used for evaluation.

Platform	Devices	
Desktop	Lenovo P50 computers with Windows 10.	
Web	Firefox browser, version 82.0.2, 64 bits.	
iOS	iPhone 5s, iPhone 6, iPhone 7.	
Android	Samsung, Sony, Motorola, and Asus.	

Table 4 contains data extracted from evaluated apps and displays a summary of 10 random apps: original MARS score, competitive classification group, and new MARS score. The competitive classification group is defined by a competitive neural network applied to MARS data. The new MARS score is the average of the most significant MARS items (X2, X5, X6, X8, X11, X15) given by a greedy stepwise algorithm.

Table 4 Data extracted from Apps evaluated.

Variables	Blindfold sudoku	Cross fingers	Dibuja el abecedario	Diferencia animales	Juego de clasificación	Memora –classic	Memora 2	Oldschool blocks	Iwritemusic	Keezy classic	
Engagement	1. Entertainment	X1	3	5	4	4	4	4	5	5	5	4	
2. Interest	X2	4	5	4	4	4	5	5	5	4	4	
3. Customization	X3	3	5	5	5	3	4	4	5	4	5	
4. Interactivity	X4	4	5	4	5	4	4	4	5	4	4	
5. Target group	X5	4	4	5	5	4	5	5	4	4	5	
Functionality	6. Performance	X6	4	5	5	5	4	5	4	5	4	4	
7. Ease of use	X7	4	5	5	5	5	5	5	4	5	5	
8. Navigation	X8	5	5	5	5	5	5	5	4	4	5	
9. Gestural design	X9	5	4	5	5	5	5	5	5	4	5	
Aesthetics	10. Layout	X10	4	5	4	4	4	4	5	5	5	5	
11. Graphics	X11	3	4	5	4	3	4	4	4	4	5	
12. Visual appeal	X12	3	4	4	4	4	4	5	4	5	4	
Information	13. Accuracy of app description	X13	4	5	5	4	4	5	4	5	5	4	
14. Goals	X14	4	5	5	5	5	4	5	5	4	4	
15. Quality of information	X15	5	4	4	4	4	5	4	5	5	5	
16. Quantity of information	X16	4	5	5	5	4	4	5	4	4	4	
17. Visual information	X17	5	5	5	5	4	4	5	4	4	4	
18. Credibility	X18	4	4	5	5	5	5	5	5	5	4	
	Original MARS quality score		4.0	4.7	4.7	4.6	4.2	4.5	4.7	4.6	4.4	4.4	
	Competitive classification group		5	4	4	4	5	5	4	4	1	3	
	New MARS quality score		4.2	4.5	4.7	4.5	4	4.8	4.5	4.5	4.2	4.7	

Interpretation of the results

Cronbach’s α is the most useful as a positive test to determine an instrument’s internal consistency (Straub & Gefen, 2004). An appropriate reliability score is one of 0.7 or higher (Heale & Twycross, 2015). In this case, the value is 0.966; but this value suggests there are data item duplications. Table 5 shows the data regression matrix corresponding to categorical variables and a high linear correlation between some of them; this result can also be related to data item duplications. The gray cells show the values which have a higher correlation between the variables; higher values are considered greater than 0.5.

Table 5 Data regression matrix.

	X1	X2	X3	X4	X5	X6	X7	X8	X9	X10	X11	X12	X13	X14	X15	X16	X17	X18	
X1	1	0.706	0.381	0.283	0.475	0.381	0.465	0.410	0.383	0.411	0.359	0.376	0.446	0.462	0.467	0.421	0.396	0.131	
X2	0.706	1	0.402	0.318	0.546	0.443	0.399	0.413	0.406	0.420	0.317	0.351	0.460	0.476	0.476	0.405	0.404	0.170	
X3	0.381	0.402	1	0.650	0.381	0.340	0.282	0.315	0.278	0.325	0.541	0.503	0.359	0.438	0.407	0.435	0.391	0.050	
X4	0.283	0.318	0.650	1	0.263	0.172	0.270	0.263	0.138	0.184	0.612	0.560	0.250	0.384	0.396	0.412	0.357	0.050	
X5	0.475	0.546	0.381	0.263	1	0.519	0.488	0.448	0.437	0.478	0.266	0.301	0.442	0.492	0.417	0.423	0.438	0.219	
X6	0.381	0.443	0.340	0.172	0.519	1	0.400	0.509	0.580	0.489	0.222	0.248	0.445	0.418	0.381	0.304	0.398	0.148	
X7	0.465	0.399	0.282	0.270	0.488	0.400	1	0.599	0.431	0.359	0.306	0.342	0.401	0.512	0.416	0.404	0.446	0.104	
X8	0.410	0.413	0.315	0.263	0.448	0.509	0.599	1	0.571	0.435	0.246	0.267	0.352	0.482	0.347	0.320	0.385	0.101	
X9	0.383	0.406	0.278	0.138	0.437	0.580	0.431	0.571	1	0.578	0.128	0.261	0.457	0.422	0.371	0.307	0.397	0.251	
X10	0.411	0.420	0.325	0.184	0.478	0.489	0.359	0.435	0.578	1	0.236	0.324	0.459	0.428	0.436	0.354	0.399	0.297	
X11	0.359	0.317	0.541	0.612	0.266	0.222	0.306	0.246	0.128	0.236	1	0.698	0.316	0.383	0.425	0.431	0.348	0.088	
X12	0.376	0.351	0.503	0.560	0.301	0.248	0.342	0.267	0.261	0.324	0.698	1	0.429	0.458	0.475	0.497	0.440	0.264	
X13	0.446	0.460	0.359	0.250	0.442	0.445	0.401	0.352	0.457	0.459	0.316	0.429	1	0.620	0.632	0.520	0.564	0.286	
X14	0.462	0.476	0.438	0.384	0.492	0.418	0.512	0.482	0.422	0.428	0.383	0.458	0.620	1	0.635	0.626	0.573	0.247	
X15	0.467	0.476	0.407	0.396	0.417	0.381	0.416	0.347	0.371	0.436	0.425	0.475	0.632	0.635	1	0.699	0.689	0.256	
X16	0.421	0.405	0.435	0.412	0.423	0.304	0.404	0.320	0.307	0.354	0.431	0.497	0.520	0.626	0.699	1	0.728	0.226	
X17	0.396	0.404	0.391	0.357	0.438	0.398	0.446	0.385	0.397	0.399	0.348	0.440	0.564	0.573	0.689	0.728	1	0.260	
X18	0.131	0.170	0.050	0.050	0.219	0.148	0.104	0.101	0.251	0.297	0.088	0.264	0.286	0.247	0.256	0.226	0.260	1	

The most significant descriptive statistical results between the non-linear distribution of the variables and the direct proportional relationships are shown in Fig. 4. The non-linear distributions are the consequence of the categorical variables of the Likert scale used in MARS. The positive proportional relationship between all variables shows that the feedback was 100% positive. This effect would be a strange result only possible in inexistent open systems, where the possible improvements are limitless.

Figure 4 Scatter plot matrix for the first nine variables.

Summarizing the previous facts: (1) The high linear correlations suggest that some variables introduce duplications in data; (2) The distribution of categorical variable values are non-linear; therefore, possible models for treating the data must support non-linear data; (3) Considering the perspective of the MARS results, the tool defines a quality value of apps that is ultimately accepted and not guide apps’ quality measurement and interpretation process satisfactorily. As MARS was systematically defined, it is insufficient to understand the ratings. Therefore, theorizing and applying a technique to reduce MARS factors is a feasible research objective to utilize abductive reasoning.

The rule: the new explanatory model

A variable is represented by a feature that is a specific quantifiable property of a procedure being observed. Feature selection assists in the comprehension of data, reducing computer skill requirement, simplifying dimensionality, and improving the predictor performance. Subsequently, the focus of feature selection is to choose a subset of input variables that can explain data, limiting the impacts from noise or superfluous variables, and still provide an improved selection of predictive results (Guyon & Elisseeff, 2003; Chandrashekar & Sahin, 2014; Saeys, Inza & Larranaga, 2007).

Label information is the feature selection technique classified into three groups: supervised methods, semi-supervised methods, and unsupervised methods (Miao & Niu, 2016; Dy & Brodley, 2004; Law, Figueiredo & Jain, 2004). Label information enables the supervised feature selection algorithms to effectively opt for discriminative and pertinent features, to highlight samples from different classes.

Feature selection is also classified into three techniques: filter, wrapper, and embedded methods (Chandrashekar & Sahin, 2014; Saeys, Inza & Larranaga, 2007; Miao & Niu, 2016). The filter models are fast and straightforward, while the embedded methods trend to performance optimization manages high data volume. The wrapper methods achieve balance.

Wrapper methods incorporate a learning algorithm, similar to a black box, and consist of utilizing the prediction performance to evaluate the relative feature of subsets of variables. Alternatively, the feature selection algorithm applies a learning method (Classifier) as a subroutine with the computational load that originates from taking a learning algorithm to assess each subset of features (Kohavi & John, 1997; Bolón-Canedo, Sánchez-Maroño & Alonso-Betanzos, 2016) (see Fig. 5).

Figure 5 Wrapper method configuration.

Adapted by permission from Springer Nature: Progress in Artificial Intelligence, Straub & Gefen (2004).

Guyon analyzed the use of criteria techniques to select features: the objective function, feature construction, feature ranking, multivariate feature selection, efficient search methods, and feature validity assessment methods (Guyon & Elisseeff, 2003). Chosen options for Feature Selection are Wrapper Subset Evaluator, Correlation-based Feature Subset Selection (CFS), Principal Components Analysis, (Abusamra, 2013; Kaur, 2016; Karabulut, Özel & Ibrikçi, 2012). Other options for Search Methods(Classifiers) are Greedy Stepwise and Best First (Chandrashekar & Sahin, 2014; Punch et al., 1993).

In this research, it is essential to recognize that the data variables are discrete and non-linear, creating limitations. Therefore, options of Feature Selection are CFS and Wrapper Subset Evaluator. Wrapper Subset Evaluator are techniques based on Bayes, Rules, Functions, and Trees, as can be studied in the works of Abusamra (2013), Kaur (2016), and Karabulut, Özel & Ibrikçi (2012).

Figure 6 illustrates the current data collection evidence that the MARS score, a mean value, is an apparent dependent variable that can be deleted without changing the data. In this case, an option is to use an unsupervised learning technique to re-classify the results. It is possible to filter the relevance of the variables utilizing a wrapper feature selection technique. At last, the software quality elements are pinpointed and interpreted.

Figure 6 The model to identify relevant quality factors.

The case

In order to apply the new model for the apps, it is essential to use one unsupervised classifier, a supervised wrapper, and a search technique. The classifier in this research study implemented a self-organizing connect with a competitive network version, able to determine consistencies and correlations in their input and adapt their future output (Ukil & Ukil, 2007).

In an initial assessment, utilizing the Merit as an efficiency measure (Hall, 1999), the optimum results were revealed by Multilayer Perceptron as a wrapper method and greedy stepwise as a search technique.

Merit is calculated as: MS=krcf¯k+kk−1rff¯

where the heuristic “Merit” of a feature subset S containing k features, rcf is the mean feature-class correlation (f ∈ S), and rff is the average feature-feature intercorrelation.

The numerator of the equation illustrates how predictive of the class a set of characteristics is, the denominator of how many of them are redundant. Consequently, the higher value of MS means better data classification.

Self-organizing neural network

The neurons of self-organizing maps learn to identify groups of comparable input vectors to ensure that neurons physically near each other in the neuron layer respond to identical input vectors (Akbari et al., 2008). The competitive learning models, a type of self-organizing maps, are based upon the principle of Winner Take All, specified as the closest weight vector to the existing input vector (Miao & Niu, 2016). The formula to discover the winning neuron, i(t), is: it= argmin∀ixt−Wit

Where x(t) is the current input vector, wi(t) is the weight vector of neuron i, and t is the iteration number. The weight vector of the winning neuron is iteratively modified, using a learning rate η(0 ≤ η ≤ 1), through Miao & Niu (2016): where x(t) is the current input vector, wi(t) is the weight vector of neuron i, and t is the iteration number. The weight vector of the winning neuron is iteratively modified, using a learning rate η(0 ≤ η ≤ 1), through Guyon & Elisseeff (2003): wit+1=wit+ηxt−wit

Greedy stepwise search method

The greedy stepwise executes a greedy forward or backward search through the area of characteristic subsets; the process finalizes when the addition/subtraction of any remaining feature causes a lesser evaluation. The method, described by Arguello (2015), solves the following model based on the Variance: maxS⊂PR2G,Ss.t.S=k

Where K is the number of data sources to choose, P is the data sources, G is the target data, and αi are the regression coefficients from fitting G using the Pi’s R2G,S=VarG−VarG−∑i∈SαiPiVarG

Multilayer perceptron

A supervised classification was building a class model from a set of records containing class labels (Lotulitr et al., 2016). A multilayer perceptron (MLP), a class of feedforward artificial neural network, categorizes data that is not linearly separable.

An MLP contains a minimum of three layers of nodes: an input layer, a concealed layer, and an outcome layer. Besides the input nodes, each node is a neuron that uses a non-linear activation feature. MLP applies a supervised learning method, backpropagation, for training. Its numerous layers and non-linear activation differentiate MLP from a linear perceptron. MLP is formalized by: y=hA=hgI=hgf∑Xpi×Wji

Where Xpi is the input vector of dimension p; f is the input function; g is the activation function, and h represents the training function. Weights Wji are updated using a backpropagation process.

Model application and results

According to the process specified in Fig. 4, the application’s details and results are documented. The MARS output data are categorized by the competitive model (see Fig. 7). Four is the logical number of categories that coincides with the number of Mars categories.

Figure 7 Competitive neural network architecture used for data classification.

Table 6 displays a summary of the classification results, the best of several attempts in each one, which is improved by trial-and-error technique until the response was stable. Different combinations of the learning rate, initial weights, and iterations are examined; in addition, the Merit value is considered, according to the explanation below. MATLAB’s nntool, with 500 epochs and a learning scale of 0.1 was applied for data processing.

Table 6 Summary of the cases classification.

Class	Number of cases	%	
1	109	19.29	
2	131	23.19	
3	65	11.50	
4	142	25.14	
5	118	20.88	

The results show that the MARS score and the competitive classification are not necessarily similar (see Fig. 8 and Table 4). This fact can be interpreted as the way the MARS tool users understand the model’s questions in a particular form for each app since the mean value assumes the same interpretation in all app cases.

Figure 8 MARS evaluation values and competitive classification.

The classified data is applied to choose the relevant variables with the greedy stepwise algorithm as a search method and J48, an extension of ID3 and C4.5, as a classifier. The combination was run as a wrapper method in Weka (Witten et al., 2011; Bouckaert et al., 2013).

The value of Merit (MS) of the feature selection (Witten et al., 2011) was 1.024. The algorithms produce as selected attributes the variables X2, X5, X6, X8, X11, X15 as relevant for the study. Using the chosen variables, it is possible to recalculate the MARS quality score (see Table 4, for example, Fig. 9). There is a similitude on the tendency of the value.

Figure 9 Old and new MARS scores of apps.

The results of competitive classification can be useful to select apps of similar quality. Each app case requires an individual analysis; for example, the competitive classification in a group of apps Blindfold sudoku and Memora –classic (See Table 4) is identical, despite the original MARS scores are different. In another case, the competitive classification can be different, although the original MARS scores are similar. The competitive trained model can also be used to classify a MARS evaluation of a new app and identify others of similar quality.

Finally, the group of six selected variables proves the following:

1. The four MARS categories are maintained, but some subcategories are optional.

2. The interest and target group represent the engagement category.

3. Performance and navigation represent the functionality category.

4. Graphics represent the aesthetics category.

5. The quality of information is defined as the information category.

Discussion

The use of the abductive process to theory generation is summarized in Fig. 10. Initially, the result has an old hypothesis: the Apps quality evaluation, using the MARS tool, facilitates the complete identification and interpretation of quality factors. In this phase, the MARS tool is used considering the practices applied in relevant research and recommended by the tool’s authors.

Figure 10 Abductive reasoning process used in this research.

The abductive reasoning process (left). The hypotheses and the application process of the new model (right).

After obtaining the MARS evaluation values, a disruption is identified from the use quality (ISO, 2014). Although the MARS tool was created using a systematic process, its application shows average values and a set of descriptive statistical values, which do not permit new explanations and interpretations about the apps’ quality factors.

As such, a new rule or model is necessary. The following is the new hypothesis: the results of apps’ quality evaluation enable selecting quality features, using data mining techniques ordered in a new processing model. In this phase, the MARS tool results are processed using a new model to obtain the relevant quality factors. A case is defined and developed; that is, the new rule is applied to the original MARS evaluation results, generating further explanations and interpretations of the old results. The new findings have useful evidence about their validity.

As is noted in the data collection, the data corresponding to one evaluation per app, and the group have a domain related to people with disabilities, which run on similar technological platforms (mobiles). These facts can be interpreted as the data capture the generalized quality of apps in the specified domain and specific users. Collaterally, the classification obtained of the competitive neural network enables to identify of the classification group and the apps with similar quality.

Feature selection process identifies six relevant variables, and according to Chandrashekar & Sahin (2014), which assists with the interpretation of data, minimizing the effect of the dimensionality, and increasing the predictor performance.

Reducing dimensionality permits a better understanding of the quality factors. In this way, the components of a quality profile for apps for people with disabilities are settings, interactions, goals, and information. This profile can be used as a criterion for apps quality improvement.

In this research, reducing the computation requirement is not an objective because of the data size. But the possibility of the use the selected variables to construct a small questionnaire directed to final users and specialists is an essential output of the new explanation. The result constitutes an improvement to the predictor performance. It can be recognized as a significant outcome, which would contribute to react adequately to the dynamics of the current context of development and the massive emergence of mobile Apps.

In many data mining machine learning applications, the precise knowledge structures are acquired, the structural descriptions are equally as important as the ability to perform well on new examples (Grainger et al., 2017). Also, researchers regularly use data mining to extract knowledge, not only predictions (Witten et al., 2011). Both opinions support the idea of a theory generation, according to the assertions of Horváth (2016) and Wacker (1998) considered in the related work of this study.

The contents of this study identify the collected data as MARS application results (what), describe (how), and explain (why) their significance, based on a new quantitative model. Similarly, the study establishes the conditions for the new model (when and where). Therefore, according to Recker (2013), cited in the related work, the experience described in this study reached the generation of an explanatory theory.

Conclusions and Future Work

The post-positivist philosophies of social science have identified the basic restrictions of the positivist behavioral approach to IS research study and present new goals for the systematic development of scientific research practice. Therefore, further research is of the utmost importance (Osei-Bryson & Ngwenyama, 2014a).

The abductive approach has been used in IS domains, as well as the multiple options to use quantitative techniques are considered and potentiate the results. The application is also related to apps quality using data mining techniques and evidence a practical use case; an initial quantitative model is analyzed using other specialized quantitative models.

The investigative community has generated several qualitative and quantitative models in varied domains, like MARS. So, the general concepts of this work are applicable because the used process constitutes an assessment of the proposed model’s external consistency (Rule); that is, according to Brown (2016) and the outer reliability enhanced/verified by inspecting statistical results regarding process replication.

The main contribution of this work is to decrease from 18 to six items of the MARS to evaluate apps; those selected attribute the variables X2, X5, X6, X8, X11, X15 as relevant to the study. This reduction in the number of variables reduces the time needed to evaluate the quality of an app since fewer items are needed, but without a decrease in the quality of the results.

Of the investigations mentioned, the evaluators are health specialists and the article’s authors. Only in the previous research, the app’s evaluation was carried out by final users, a cancer survivors’ group. A research opportunity exists to expand the coverage of assessment, considering the users with disabilities.

In the present research, it is possible to stimulate suggestions for improvement and study the validity of generalizations, starting with machine learning for data mining.

The experience results contribute to new quantitative possibilities, such as using other intelligent options and multivariate statistical techniques to identify factors of new domains, not necessarily including feature selection. Also, simulation models can be utilized to experiment with various scenarios and identifying transcendental quality factors.

A pending research theme related to the experience presented here is the stakeholders’ participation in the apps evaluation process. Based on the preliminary evidence described, the results of this research are invaluable. Additionally, comparative studies could be beneficial for the final user and specialists’ involvement.

With this research, academics can revise a new experience using an alternative reasoning process to overcome IS research’s positivism. For the practitioners, the study contributes to the growth of the current knowledge about apps quality assessment related to people with disabilities.

Supplemental Information

Supplemental Information 1 Data of evaluation of Apps for People with Disabilities using the Mobile App Rating Scale (MARS) tool

Click here for additional data file.

Supplemental Information 2 Questionnaire MARS Spanish

Click here for additional data file.

Supplemental Information 3 Questionnaire MARS

Click here for additional data file.

Additional Information and Declarations

Competing Interests

Author Contributions

Data Availability

The authors declare there are no competing interests.

Andres Larco, Carlos Montenegro and Cesar Yanez conceived and designed the experiments, performed the experiments, analyzed the data, prepared figures and/or tables, authored or reviewed drafts of the paper, and approved the final draft.

Sergio Luján-Mora conceived and designed the experiments, performed the experiments, prepared figures and/or tables, authored or reviewed drafts of the paper, and approved the final draft.

The following information was supplied regarding data availability:

The data of the evaluation of Apps for People with Disabilities using the Mobile App Rating Scale (MARS) tool are available in the Supplementary File.

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
