# Peer review of "An experience selecting quality features of apps for people with disabilities using abductive approach to explanatory theory generation"

_PeerJ Computer Science, doi:10.7717/peerj-cs.595_

## Round 0.1 · original submission · Minor Revisions

· Academic Editor

Minor Revisions

The paper should be improved in terms of presentation and structure.

Reviewer 1 ·

Basic reporting

The authors have explained very well first 3 sections. There is need of addition in section 2 that how the proposed work is different with existing one.

Experimental design

All the results are mentioned in an effective manner.

Validity of the findings

NIL

Additional comments

Acceptable

Reviewer 2 ·

Basic reporting

This paper "An experience selecting quality features of apps for
people with disabilities using abductive approach to
explanatory theory generation" is well organized and reasonable in structure. The comments are as follows:

Does the paper contribute to the body of knowledge?: Yes.

The paper proposes the first attempt
to improve the MARS tool, aiming to provide specialists relevant data, reducing noise
effects, accomplishing better predictive results to enhance their investigations.

this methodology allows the developers to use to analyze and develop software for the health informatics model and create a space in which software engineering and machine learning experts can work together on the machine learning model lifecycle.

Is the paper technically sound?: Yes. The paper is technically sound and is of very high quality. The various claims in the paper are quite well supported by the experiments and evaluate on a set of real data sets. the manuscript presents many FIGURE and many tables, this supports the researches idea and methodology. The results on real-world data sets are promising and motivate further investigations into the use of approximate inference in this context.

Is the subject matter presented in a comprehensive manner?: no. The presentation isn't comprehensive it needs to better organize and simplify the main objectives of the idea of the manuscript topic

Are the references provided applicable and sufficient?: Yes. Among all references, only poor references are in the past three years. In order to highlight the innovation of this work, it is better to cite other six up-to-date references to be applicable and sufficient enough to provide relevant materials about this novel approach software.

Experimental design

This paper "An experience selecting quality features of apps for
people with disabilities using abductive approach to
explanatory theory generation" is well organized and reasonable in structure. The comments are as follows:

Does the paper contribute to the body of knowledge?: Yes.

The paper proposes the first attempt
to improve the MARS tool, aiming to provide specialists relevant data, reducing noise
effects, accomplishing better predictive results to enhance their investigations.

this methodology allows the developers to use to analyze and develop software for the health informatics model and create a space in which software engineering and machine learning experts can work together on the machine learning model lifecycle.

Is the paper technically sound?: Yes. The paper is technically sound and is of very high quality. The various claims in the paper are quite well supported by the experiments and evaluate on a set of real data sets. the manuscript presents many FIGURE and many tables, this supports the researches idea and methodology. The results on real-world data sets are promising and motivate further investigations into the use of approximate inference in this context.

Is the subject matter presented in a comprehensive manner?: no. The presentation isn't comprehensive it needs to better organize and simplify the main objectives of the idea of the manuscript topic

Are the references provided applicable and sufficient?: Yes. Among all references, only poor references are in the past three years. In order to highlight the innovation of this work, it is better to cite other six up-to-date references to be applicable and sufficient enough to provide relevant materials about this novel approach software.

Validity of the findings

This paper "An experience selecting quality features of apps for
people with disabilities using abductive approach to
explanatory theory generation" is well organized and reasonable in structure. The comments are as follows:

Does the paper contribute to the body of knowledge?: Yes.

The paper proposes the first attempt
to improve the MARS tool, aiming to provide specialists relevant data, reducing noise
effects, accomplishing better predictive results to enhance their investigations.

this methodology allows the developers to use to analyze and develop software for the health informatics model and create a space in which software engineering and machine learning experts can work together on the machine learning model lifecycle.

Is the paper technically sound?: Yes. The paper is technically sound and is of very high quality. The various claims in the paper are quite well supported by the experiments and evaluate on a set of real data sets. the manuscript presents many FIGURE and many tables, this supports the researches idea and methodology. The results on real-world data sets are promising and motivate further investigations into the use of approximate inference in this context.

Is the subject matter presented in a comprehensive manner?: no. The presentation isn't comprehensive it needs to better organize and simplify the main objectives of the idea of the manuscript topic

Are the references provided applicable and sufficient?: Yes. Among all references, only poor references are in the past three years. In order to highlight the innovation of this work, it is better to cite other six up-to-date references to be applicable and sufficient enough to provide relevant materials about this novel approach software.

Additional comments

This paper "An experience selecting quality features of apps for
people with disabilities using abductive approach to
explanatory theory generation" is well organized and reasonable in structure. The comments are as follows:

Does the paper contribute to the body of knowledge?: Yes.

The paper proposes the first attempt
to improve the MARS tool, aiming to provide specialists relevant data, reducing noise
effects, accomplishing better predictive results to enhance their investigations.

this methodology allows the developers to use to analyze and develop software for the health informatics model and create a space in which software engineering and machine learning experts can work together on the machine learning model lifecycle.

Is the paper technically sound?: Yes. The paper is technically sound and is of very high quality. The various claims in the paper are quite well supported by the experiments and evaluate on a set of real data sets. the manuscript presents many FIGURE and many tables, this supports the researches idea and methodology. The results on real-world data sets are promising and motivate further investigations into the use of approximate inference in this context.

Is the subject matter presented in a comprehensive manner?: no. The presentation isn't comprehensive it needs to better organize and simplify the main objectives of the idea of the manuscript topic

Are the references provided applicable and sufficient?: Yes. Among all references, only poor references are in the past three years. In order to highlight the innovation of this work, it is better to cite other six up-to-date references to be applicable and sufficient enough to provide relevant materials about this novel approach software.

Reviewer 3 ·

Basic reporting

In this paper the authors present an study that determines the most relevant quality factors of apps for people with disabilities utilizing the abductive approach to the generation of an explanatory theory. The paper reads well and it is well organized. Very few recent references are included.

Experimental design

The contributions of the paper and the research questions should be clarified.

Validity of the findings

The results are well presented and support the findings presented in the discussion and conclusions sections.

Additional comments

Some aspects to improve:
- Introduction: clarify the contributions and research questions.
- Related Work: Analyze the references more deeply and include more references of the latest 5 years.

---

## Round 0.2 · accepted · Accept

· Academic Editor

Accept

The paper can be accepted

Reviewer 2 ·

Basic reporting

The manuscript is the best one that can contribute to scientific research. I recommend acceptance to publish in the peerj journal, according to the instructions and conditions of publication of peerj journal.

Experimental design

All results are displayed intact and clearly

Validity of the findings

Perfect to an extent

Additional comments

Acceptable

Reviewer 3 ·

Basic reporting

The authors have addressed all my concerns.

Experimental design

The authors have addressed all my concerns.

Validity of the findings

The authors have addressed all my concerns.

Additional comments

The authors have addressed all my concerns.